# Assessment of the Microbiological Quality of Ready-to-Eat Salads—Are There Any Reasons for Concern about Public Health?

**DOI:** 10.3390/ijerph19031582

**Published:** 2022-01-29

**Authors:** Anna Łepecka, Dorota Zielińska, Piotr Szymański, Izabela Buras, Danuta Kołożyn-Krajewska

**Affiliations:** 1Department of Meat and Fat Technology, Prof. Waclaw Dabrowski Institute of Agriculture and Food Biotechnology—State Research Institute, 02-532 Warsaw, Poland; piotr.szymanski@ibprs.pl; 2Department of Food Gastronomy and Food Hygiene, Institute of Human Nutrition Sciences, Warsaw University of Life Sciences-SGGW, 02-776 Warsaw, Poland; dorota_zielinska@sggw.edu.pl (D.Z.); izabelaburas94@gmail.com (I.B.); danuta_kolozyn_krajewska@sggw.edu.pl (D.K.-K.)

**Keywords:** RTE, microbial, evaluation, *Salmonella*, *Listeria monocytogenes*

## Abstract

Ready-to-eat food products can be readily consumed without further preparation and are convenient for busy on-the-go consumers. The objective of the study was to assess the microbiological quality of ready-to-eat salads. Thirty RTE salads were tested for the presence of bacteria, yeasts, and molds using the TEMPO and agar plate method. The study demonstrated that most of the tested products were characterized by varying microbiological quality. The total number of mesophilic microbiotas was about 6 log CFU g^−1^. The high number of microorganisms was due to yeast and molds or *Enterobacteriaceae*. Half of the salads were contaminated with *E. coli* and three salads were contaminated with *S. aureus*. LAB were also found, which can be explained mainly by a dairy ingredient. In some salads, *Salmonella* spp. and *L. monocytogenes* were detected (26.7% and 33.3% of the samples, respectively). Based on the conducted tests, it was found that the microbiological quality was not satisfactory. The results presented in this study indicate that there is a significant problem of the presence of pathogens. Manufacturers should strive to reduce the possibility of microbial contamination through the use of widely understood hygiene of the production process, using hurdle technology, including the modified atmosphere and refrigerated storage.

## 1. Introduction

Convenience food has become an alternative to traditionally prepared food. Convenience food can be defined as: food, typically a complete meal, that has been pre-prepared commercially and so requires minimum further preparation by the consumer. The market offers more and more new high-quality food products, produced using new recipes, modern production and packaging technologies [1,2]. 

Currently, the demand for ready-to-eat (RTE) products, including salads, is increasing. RTE salads are minimally processed products. By definition, minimally processed food has the appearance of fresh food, is characterized by the least changed characteristics, and is safe to eat. RTE food products can be readily consumed without further preparation and are convenient for busy consumers. The production of ready-to-eat salads is constantly growing and they are currently produced on an industrial scale [3]. The availability of both RTE fruit and vegetables on the market may contribute to increasing the consumption of fruit and vegetables in the general population and thus contribute to efforts to achieve a daily consumption of 400 g of vegetables and fruit per capita, as recommended by the World Health Organization [4]. It was estimated that in Europe in 2021, the average amount of ready-to-eat meals consumed per person was about 15 kg [5].

In addition to providing the necessary ingredients, food items should be characterized by health quality, including appropriate microbiological quality, guaranteeing safety to health. Healthy food quality depends on the conditions and ways of obtaining the raw materials, pre-treatment and thermal processing, storage, transport, and conditions of sale [3]. All these operations can result in damage to the natural barrier of the epidermis and disrupt the internal compartmentalization, which in turn separates enzymes from substrates, thus, promoting microbial proliferation and inducing increased respiration in the plant. This, in turn, results in an increase in metabolism and the aging of tissues [4]. Due to numerous cases of food poisoning caused by microorganisms and toxins found in food, research and the control of food products are important. There are legal regulations regarding food, defining microbiological criteria and limits for the occurrence of microorganisms in food. These criteria, however, apply to selected food groups and regulate the selected acceptable limits of microorganisms [6,7].

According to the WHO [8], 550 million people suffer from *Salmonella* poisoning every year, and about 220 million of them are children under 5 years of age. *Salmonella* is one of the four key global causes of diarrheal diseases. Currently, in the European Union, salmonellosis is the second most deadly disease, after campylobacteriosis, causing food poisoning [9] with a notification rate of 20.0 cases per 100,000 population [10]. The trend towards human salmonellosis has been stable over the past five years after a long period of a downward trend. Additionally, *Salmonella* Enteritidis caused the vast majority (72.4%) of food-borne salmonellosis outbreaks [10].

Subsequently, 1 million people suffer from listeriosis per year [11]. Unfortunately, most cases require hospitalization. According to EFSA and ECDC [9], the number of poisonings by *Listeria monocytogenes* in the European Union is constantly increasing. The listeriosis notification rate is 0.47 cases per 100,000 people [12]. One of the reasons for this increase is the interest in ready-to-eat products. Based on the EFSA and ECDC study [9], *L. monocytogenes* was found in RTE foods in the largest number of samples of fish and fish products (6.0% of samples), salads (4.2% of samples), meat and meat products (1.8% of samples), soft and semi-soft cheeses (0.9% of samples), fruit and vegetables (0.6% of samples), and hard cheeses (0.1% of samples). According to the EFSA BIOHAZ Report [13] ready-to-eat salads were found to be the main source of poisoning by *L. monocytogenes*. In 2013, in Germany, three human cases of listeriosis were found after eating ready-to-eat vegetables, juices, and mixed salads (one death was reported). In the same country, in 2014, two cases were observed after eating mixed food (iceberg lettuce with yogurt dressing, and gouda cheese). In Switzerland, thirty-one human listeriosis cases were found in 2014 (four deaths were reported). *L. monocytogenes* were found in ready-to-eat vegetables, juices and pre-cut salads. 

Ready-to-eat salads could show the presence of various microbial pathogens including *Escherichia coli*, *Staphylococcus aureus*, *Salmonella* spp., *Listeria monocytogenes*, *Campylobacter jejuni*, *Clostridium perfringens*, total aerobic and spoilage bacteria, yeasts, and molds, which are concerned with serious threats [14,15]. In addition, a high number of microorganisms are also becoming a reservoir of antibiotic resistance genes, which has now become a global problem [16,17].

Although the European Commission Regulation No. 852/2004 [18] on the hygiene of foodstuffs requires businesses to implement Good Hygiene Practice and a food safety management system based on hazard analysis and critical control point (HACCP) principles, many authors indicated that RTE food causes high microbiological risk [19]. However, the knowledge of factors affecting the quality of RTE foods is still inconclusive.

The main objective of this study was to assess the microbiological quality of ready-to-eat salads from the Polish market and to evaluate whether the composition, time, and method of packaging had an impact on the shelf life of these salads.

## 2. Materials and Methods

### 2.1. Materials

Thirty salads were purchased in autumn at several discount stores from the Polish market (Warsaw) and retail chains. The products were fresh, before the expiry date, transported in thermal bags, and then refrigerated (4–8 °C) until the beginning of the study. The samples were tested on the day of purchase. All of the products were tightly packed in plastic boxes like ‘lunch boxes’. Three salads of each type were purchased from the same production batch and tested. Table 1 and Table 2 present information about the tested salads according to the manufacturers’ declarations.

The mixed ingredients of the ready-to-eat salads included raw and cooked ingredients, e.g., vegetables (leafy vegetables, cherry tomatoes, and carrots), meat (ham, smoked chicken, and grilled chicken), fish (salmon and tuna), cheeses, and carbohydrate sources (pasta, and buckwheat). The tested salads differed in composition, shelf-life, the way they were packed and storage temperature. The recommended maximum storage temperature as indicated on the label was 10 °C. The shelf life was 1–7 days. Some of the salads had dressings/sauces in sachets and some of the salads were mixed and ready to eat. In most cases, the producers did not give the composition of the dressings/sauces. Some of the dressings/sauces in the salads were with the addition of preservatives.

### 2.2. Methods

#### 2.2.1. Samples Preparation

Three salads were prepared from each production batch. First, 25 g of the salad product was weighed by taking different ingredients to obtain a representative sample. Then, 225 mL of peptone water (BioMaxima, Lublin, Poland) was added and homogenized in a stomacher (Stomacher 80 Biomaster, Seward Limited, London, UK) for 5 min. Filtered stomacher bags (BagFilter^®^ 400 P, Interscience, Paris, France) were used to eliminate any solid particles. Decimal dilutions were made, and the prepared samples were used for further testing.

#### 2.2.2. Microbiological Analysis

Microbiological tests (count of aerobic mesophilic total flora, *Staphylococcus aureus*, *Enterobacteriaceae*, *Escherichia coli*, lactic acid bacteria, yeasts, and molds) were carried out using the TEMPO^®^ system (bioMérieux, Marcy-l’Étoile, France). The dehydrated culture media in the bottles were prepared by adding 3.9 mL of sterile distilled water. Then, 0.1 mL of the sample solution was added to them in a suitable dilution and vortexed (5–10 s; Heidolph, Schwabach, Germany). The samples were incubated in INE 400 Incubator (Memmert GmbH + Co.KG, Buechenbach, Germany) according to the manufacturer’s manual. The results are presented on a logarithmic scale (log CFU g^−1^) and standard deviation.

The samples were analyzed for the presence of microorganisms commonly isolated from RTE vegetables, i.e., *Salmonella* spp. and *L. monocytogenes*. The plate method was used according to ISO standards [20,21]. 

The samples of the salads (25 g) were mixed with peptone water (BioMaxima, Lublin, Poland) and incubated at 37 °C for 24 h to pre-incubation. For selective propagation, nutrient broth (BioMaxima, Lublin, Poland) was used. An aliquot of 1 mL was withdrawn and transferred to 9 mL of broth and incubated again at 37 °C for 24 h. Sterile Petri dishes were poured into 15–20 mL of BGA (Brilliant Green Agar; Oxoid, Waltham, MA, USA) and, after setting with an automatic pipette, surface culture was performed. Subsequently, 0.1 mL of the sample was applied to the surface of the substrate and spread over the substrate with a sterile paddle. The cultures were incubated at 35 °C for 24 h. In the case of a positive result on the BGA, a confirmation on the XLD agar was made (Xylose Lysine Deoxycholate agar; LabM, Heywood, UK). Agar was poured onto a sterile plate and the surface was set after solidification. Subsequently, 1 mL of the inoculated nutrient broth was applied to the surface of the medium and spread over the substrate. The plates were incubated at 37 °C for 24 h. The results are presented as the presence (+) or absence (−) of *Salmonella* spp. 

Samples of the salads (25 g) were mixed with half-Fraser broth (Oxoid, Waltham, MA, USA) and incubated at 37 °C for 24 h. Then, 1 mL was removed and transferred to the Fraser broth (Oxoid, Waltham, MA, USA) and incubated for 24–48 h. In sequence, 15–20 mL of ALOA (Agar *Listeria* according to Ottaviani and Agosti; LabM, UK) was poured onto sterile plates and the surface was set after setting. Using an automated pipette, a 1 mL test sample was applied to the surface of the substrate and spread evenly over the substrate using a sterile paddle. The cultures were incubated at 37 °C for 24 h. In the case of a positive result on the ALOA, a confirmation on the PALCAM agar was made (LabM, Heywood, UK). Agar was poured onto a sterile plate and the surface was set after solidification. Subsequently, 1 mL of the inoculated Fraser broth was applied to the surface of the medium and spread evenly over the substrate. The plates were incubated at 37 °C for 24 h. The results are presented as the presence (+) or absence (−) of *Listeria monocytogenes*. 

#### 2.2.3. Statistical Analysis

Microbiological tests were performed in three replications (three repetitions with three different products from the same batch). All of the data were analyzed using the Statistica 13 (TIBCO Software Inc., Palo Alto, CA, USA). Correlation coefficients were calculated and a principal components analysis (PCA) using a correlation matrix and a cluster analysis was performed

## 3. Results and Discussion

The tested salads were characterized by different microbiological quality. Table 3 below shows the results of the number of selected microorganisms and the results of the presence of *Salmonella* spp. and *L. monocytogenes*.

There are not many studies on microbiological tests on salads with dressings/sauces. Typically, the studies apply only to raw vegetables or salad blends [22,23,24,25,26,27,28], only dressings/sauces and pesto [29], or RTE products in general [30]. These mixtures are the least processed and form a large part of the product, but they are not the only factors that determine the quality of the final product. When ingredients are mixed together with green leaves, cross-contamination may occur at any point in the production chain to consumption. Cross-contamination can occur during processing when the equipment in contact with potentially contaminated products is not regularly sanitized and cleaned [31]. 

Although there are no established microbiological criteria for ready-to-eat foods in the European Union, the only applicable regulation is Commission Regulation 1441/2007 (formerly Commission Regulation 2073/2005) [6,7]. However, it does not include this category of food, but only their individual components. For fruit and vegetables, the limit is established for bacteria *E. coli* (1000 CFU g^−1^ of product) and precut ready-to-eat fruit and vegetables are limited in *Salmonella* (absence in 25 g of product). Ready-to-eat meat products are limited in *L. monocytogenes* (100 CFU g^−1^ of a product or absence in 25 g of a product) [7,32].

Although the total viable count is not a legislation criterion for RTE salads, it is an important hygienic and sensory quality indicator, which may inform about the total microbiological status of the food. In the present study the total number of aerobic mesophilic microorganisms found in the tested salads were on a different level and, on average, about 6 log CFU g^−1^ (ranged from 2.36 log CFU g^−1^ to 9.30 log CFU g^−1^) (Table 3). The lowest total viable count of microorganisms was evaluated in salad S4. Salads S8, S9, S10, S11, S16, S19, and S23 were characterized by a high number of total viable counts. The high numbers of organisms in S16 and S23 salads were due to the high number of yeast and molds (7.00 log CFU g^−1^ and 6.60 log CFU g^−1^, respectively) or *Enterobacteriaceae* family bacteria (6.84 log CFU g^−1^). In salads S16, S19, and S23, high numbers of lactic acid bacteria were found. The count of total microbiota in the salads in this study was similar to those observed in Polish studies by Berthold-Pluta et al. [33] who reported that the count of aerobic mesophilic total microbiota in leafy vegetables and their mixes ranged from 5.6 log CFU g^−1^ to 7.6 log CFU g^−1^. Similarly, Jeddi et al. [34] reported that the total mesophilic microbiota observed in vegetables from Iran ranged from 5.3 log CFU g^−1^ to 8.5 log CFU g^−1^. Importantly, the count of aerobic mesophilic microbiota is an indicator of only the overall microbiological quality of a food product and that there are no binding standards for the quality of products of this type [33]. In general, aerophilic microbiota is capable of growing, even at low temperatures; hence, its high number even when products are stored in a refrigerator. The high number of packed ready-to-eat salads of all types of vegetables in Portugal was classified as unsatisfactory due to the presence of more than 6 log CFU g^−1^ aerobic mesophilic microorganisms, even if *Salmonella* and *L. monocytogenes* were not detected in any ready-to-eat salad samples [35]. Adopting only this criterion in our research, we should classify 18 out of 30 of the tested salads as being unsatisfactory.

In three of tested salads (S7, S17, and S18) the *S. aureus* bacteria (2.00 log CFU g^−1^, 3.54 log CFU g^−1^, and 2.9 log CFU g^−1^, respectively) (Table 3) was revealed. In seven samples of salads (S7, S9, S10, S13, S14, S15, and S23), bacteria of the *Enterobacteriaceae* family in a number higher than 6 log CFU g^−1^ were found, which indicates a high degree of microbiological contamination. Leff and Fierer [36] reported that vegetables, e.g., spinach and lettuce, were mainly affected by the *Enterobacteriaceae* family and half of the salads were contaminated with *E. coli*. Salads S6, S8, S14, and S23 showed high contamination levels with *E. coli* (more than 1000 CFU g^−1^ limited according to the Commission Regulation 1441/2007). In the study of Faour-Klingbeil et al. [37], in vegetable salads, a high number of bacteria *Staphylococcus* spp. (1.83–7.76 log CFU g^−1^) and bacteria from the *E. coli* group (0.33–7.15 log CFU g^−1^) were found, which indicates the possibility of the large contamination of vegetable salads with these bacteria. De Oliveira et al. [38] reported high contamination vegetable salads with *E. coli* (53.1% of tested samples). As the authors emphasize, the determination of *E. coli* is a good indicator for fecal infections, referring to fresh, cut leafy vegetables. Other authors have reported that almost all of the salad samples in Ghana were contaminated with *E. coli* and *Bacillus cereus* bacteria (96.7% and 93.3%, respectively) [39]. 

Lactic acid bacteria (LAB) were also found in the tested salads, which can be explained mainly by the presence of dairy additives as a salad ingredient, e.g., yogurt, cheese, blue cheese, and mozzarella (salads S1, S3, S5, S6, S7, S8, S9, S14, S19, S23, and S25), or pickled products like pickled cucumber (salads S16 and S18) (Table 3). LAB are a natural vegetable microbiota [40], but may also contaminate a product [41,42]. Some LAB were found in fresh-cut vegetables (like iceberg, lettuce, or endive). A high number of LAB also affects the total number of bacteria, greatly overstating them. In the majority of the samples tested, a significant of yeast and mold was found (1.00–7.00 log CFU g^−1^). Significant yeast and mold contamination of food products of this type was also determined by Abadias et al. [43]. The observed numbers of yeasts and molds were lower than bacteria. Furthermore, the ranges in fresh-cut vegetables were from 2.0 log CFU g^−1^ to 7.8 log CFU g^−1^. 

Fresh plant-origin products may be a vehicle for the transmission of bacterial pathogens, e.g., *E. coli*, *S. aureus, L. monocytogenes*, and *Salmonella* spp. Food contaminated by fecal microorganisms may be a source of antibiotic resistant organisms that can cause infections in people. Raw vegetables that are particularly vulnerable to contamination with bacteria are: lettuce, spinach, cabbage, cauliflower, celery, broccoli, and all salads packaged in a modified atmosphere [44]. Among animal products, poultry, meat, and eggs are the main infection sources of *Salmonella* [45,46].

The microbiological criterion for *Salmonella* spp. and *L. monocytogenes* in freshly-cut vegetables is an absence in 25 g of food [7]. In the salad samples S2, S5, S11, S14, S18, S19, S20, S21, S23, S25, S26, S27, S29, and S30, no pathogenic *Salmonella* and *L. monocytogenes* were found (46.7% of samples). *Salmonella* spp. were found in eight of the tested salads (salads S1, S3, S4, S6, S8, S16, S17, and S24). The *L. monocytogenes* species were detected in salads S7, S9, S10, S12, S13, S15, S16, S17, S22, and S28 (Table 3). The presence of these indicates high microbiological contamination and may be the cause of being affected by one of the diseases, such as salmonellosis or listeriosis. In the study of Abadias et al. [43] *Salmonella* strains were detected in corn salad, lettuce, spinach, and mixed salads (1.7% of samples were contaminated). In other studies, the occurrence of *Salmonella* in RTE vegetables varies, but usually does not exceed more than a few percent [38].

Bacteria in the *Listeria* genus are found in a variety of products, and they are clearly evident in minimal processed food. *Listeria* is a bacterium with a broad spectrum of development, capable of growing in harsh environmental conditions, and has the ability to create biofilms which may be the cause of cross-contamination [47,48,49]. According to the meta-analysis provided by Churchill et al. [50], the summary estimate of the prevalence of *L. monocytogenes* was 2.0% in packaged salads. In the study of Söderqvist et al. [26], *L. monocytogenes* was isolated from 1.4% of all RTE tested salads. This is a much lower percentage of contamination compared to our own research, as well as the EFSA and ECDC results (13.8%) [9]. According to the Scientific Report of EFSA [51], in 2010–2011, ready-to-eat samples were contaminated with *L. monocytogenes* (1.7%, 0.43%, and 0.06% for fish, meat, and cheese samples, respectively). Gurler et al. [52] found *L. monocytogenes* and *Salmonella* spp. Contamination in RTE foods commercialized in Turkey (6% and 8%, respectively). RTE meat products can be contaminated during or after processing by *L. monocytogenes* and *Salmonella* spp. [32]. Moreover, all of the *Salmonella* spp. And *L. monocytogenes* isolates exhibited resistance to one or more of the antimicrobial agents used. The results indicate the need to improve hygiene standards and implement regulations in the RTE food chain in order to ensure microbiological safety. On the other hand, Koseki et al. [53] presented data about iceberg lettuce in Japan. No pathogenic bacteria, i.e., *Salmonella*, *E. coli* O157:H7 and *L. monocytogenes*, were found. The results of the study could be used to develop risk management policies. In similar results, no pathogenic *Salmonella* in 233 vegetables, freshly-cut fruits and sprout samples, were detected by Althaus et al. [54]. Xylia et al. [19] reported that RTE salads from the Cypriot market are free from *S. enterica* and *L. monocytogenes*.

The principal components analysis (PCA) was used to analyze data obtained in the present study and revealed 20 factors that determine the quality of salads, where the first two explain 37.37% of variable variances. A dispersion of the first two factors (PC1 and PC2) on the surface is shown below in Figure 1 and Figure 2.

It was found that the modified atmosphere of package (samples S1, S2, S3, S4, S5, and S6) and preservatives (samples S2, S4, S5, and S14) used in the tested salads were negatively correlated with total aerobic microbiota (−0.46 and −0.42, respectively), as well as with *L. monocytogenes* presence (−0.35 and −0.28 respectively). Milk ingredients used in salads were correlated with lactic acid bacteria occurrence (0.53), whereas meat (S17) and eggs (S18) were correlated with *S. aureus* presence (0.31 and 0.37, respectively). On the other hand, salt added to salads was important to prevent *Salmonella* and yeast and molds developing (−0.45 and −0.33, respectively). 

The number of bacterial cells may indicate the contamination of a product, its degree of deterioration, but also may be part of the natural microbiota of the food product. According to FAO/WHO [55], leafy vegetables (spinach, cabbage, lettuce, and watercress) and fresh herbs (cilantro, basil, chicory, and parsley) are a group with a very high microbiological risk. The threat is mainly *E. coli*, *S. enterica*, *Campylobacter*, *Shigella* spp., Hepatitis A virus, Noroviruses, *Cyclospora cayetanensis*, *Cryptosporidium*, *Yersinia pseudotuberculosis*, and *L. monocytogenes*. Leafy vegetables, as a rule, cannot be subjected to thermal treatment, which prevents the deactivation of microorganisms. The cleaning of leaves is a crucial stage in the production process. Moreover, other factors can extend the shelf-life of a product, especially used as hurdle technology. 

Increased hygiene at every stage of the production process, application of GHP, GMP, and HACCP principles in the production plant, as well as maintenance of the refrigeration sequence in product storage, can increase the microbiological safety of RTE products [3,56,57]. Furthermore, packaging is a key element in the production of ready-to-eat salads, which was found in our study. Samples in which a modified atmosphere was used had lower total aerobic bacteria, as well as the *L. monocytogenes* count; however the presence of *Salmonella* spp. was higher. These results being somehow inconsistent, may be a basis for further in-depth research, including more research samples. Packing in a modified atmosphere, which involves the use of a composition of non-atmospheric gases inside the package and the use of appropriate packaging made from permeable materials, was found as an effective method [24,58,59,60]. The shelf-life of pre-packed salads is determined by microbial and chemical changes. According to Mir et al. [61] commonly used techniques for the shelf-life prolongation of RTE foods are sanitizers, modified atmosphere packaging, refrigeration, irradiation, high pressure processing, and essential oils. 

In ensuring the microbiological safety of products, it is also essential to maintain an appropriate storage temperature [28]. A low temperature, usually kept at 0–4 °C, inhibits the biochemical and chemical processes of microorganisms, which inhibits their growth in the food product [61]. Söderqvist et al. [26] recommended a temperature lower than 4 °C, while the salads tested in this study had the recommended temperature by the manufacturer even up to 10 °C. A low temperature is necessary to prevent the growth of psychrotrophs (like *L. monocytogenes*) [62]. Ziegler et al. [63] recommended a temperature lower than 5 °C, which could help minimize the risk of *L. monocytogenes* in RTE salads. Xylia et al. [19] suggest that shelf-life testing is essential to understanding and developing novel techniques to monitor the safety and quality of ready-to-eat products.

## 4. Conclusions

Based on the conducted tests, it was found that the microbiological quality of the evaluated ready-to-eat salads was not satisfactory from the safety point of view. Due to the increased interest of consumers in vegetable salad mixes with meat, fish, or cheese, carbohydrate additives (e.g., pasta and toast), and the dressings/sauces available on the market as ready-to-eat products, it is very important to study their microbiological quality. These products are minimally processed; the risk of contamination with microorganisms, including pathogenic ones, is high. 

The results presented in this study indicate that there is a significant problem of the presence of pathogenic microorganisms, mainly *L.*
*monocytogenes* and *Salmonella* sp. in ready-to-eat salads. Although no negative visual changes of the products were observed, there were a high number of bacteria, yeast, and molds in the products. It is noteworthy that although the products appear edible according to visual inspection, they often contain microorganisms that cause product spoilage, because the first signs of product spoilage may not be visible. Taking into account the limitation of the study, which was the number of samples, future investigation should include more research samples, differentiated in terms of the packaging method and season. Such data would provide valuable information and are in the great interest both of legislators and producers of food products. 

It can be summarized that RTE food manufacturers should strive to reduce the possibility of microbial contamination, through the use of widely understood hygiene production guidelines, using hurdle technology, which includes the modified atmosphere and storage of products, especially with a temperature below 5 °C. Where possible, the heat treatment of raw ingredients should be carried out, and raw products, i.e., leafy vegetables, should be thoroughly subjected to washing and drying processes.

## Figures and Tables

**Figure 1 ijerph-19-01582-f001:**
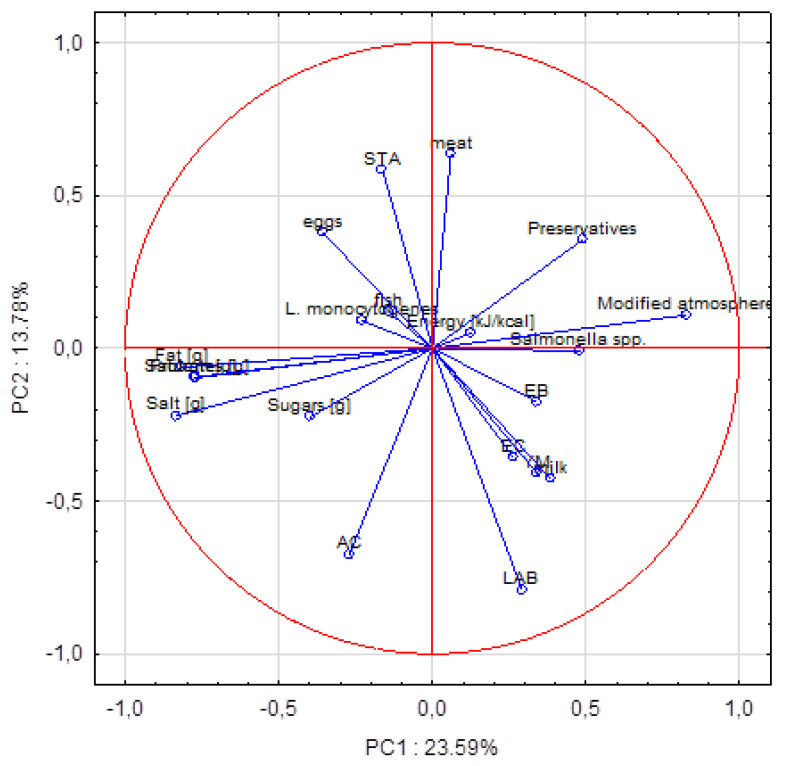
Dispersion of quality factors on a surface for the first two principal components (PC). Explanatory: AC—aerobic mesophilic total flora, STA—*S. aureus*, EB—*Enterobacteriaceae* family, EC—*E. coli*, LAB—lactic acid bacteria, and YM—yeasts and molds.

**Figure 2 ijerph-19-01582-f002:**
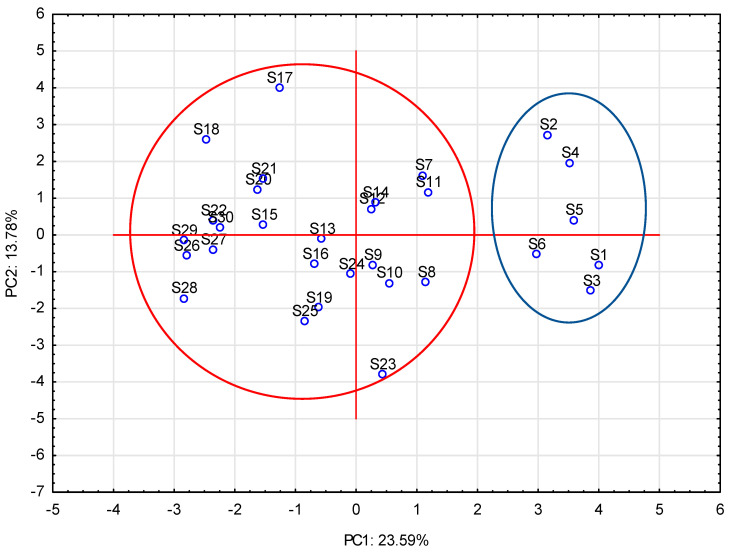
Projection of samples on a surface for the first two principal components (PC).

**Table 1 ijerph-19-01582-t001:** Ingredients of the tested salads according to the producer’s declaration.

Symbol and Name	Ingredients of Vegetable Origin	Ingredients of Animal Origin
S1—Mediterranean	Iceberg lettuce, beet leaves, cherry tomatoes, olive oil, black olives	Mediterranean cheese
S2—Smoked chicken	Lettuce, cherry tomatoes, dressing, toast	Smoked chicken
S3—Mozzarella	Lettuce, chicory, arugula, cocktail tomatoes, olive oil, toast, spices, sunflower oil, salt	Mozzarella cheese
S4—Vegetable salmon	Lettuce, cherry tomatoes	Smoked salmon, yoghurt
S5—Vegetable with blue cheese	Lettuce, radicchio lettuce, pumpkin seeds, cranberry sauce, cranberry, beetroot concentrate	Blue cheese
S6—Vegetable with mozzarella	Lettuce, chicory, radicchio, rocket, red pepper, fennel, garlic, dried tomatoes, sauce	Unripened rennet cheese
S7—With eggs and croutons	Cabbage, croutons, cherry tomatoes, salt, pepper, sugar, oregano, garlic	Boiled egg, ham, yoghurt, cream, mayonnaise
S8—Sicilian lunch	Cabbage, carrot, pepper, cucumber, sweet corn, red beans, olives, spices	Feta, yogurt, mayonnaise
S9—Caribbean lunch	Cabbage, peach, raisins, sunflower seeds, pumpkin seeds, corn, carrots, spices, soy sauce, pineapple, garlic	Mayonnaise sauce, natural yoghurt
S10—Italian lunch	Cabbage, lettuce, radish, cucumber, olives, toast, carrots, oil, spices	Cheese
S11—Pollo penne	Pasta, lettuce, cucumber, pepper, onion, spinach, capers, red cabbage	Roast chicken, sauce
S12—Gyros lunch	Lettuce, corn, cucumber, red cabbage, pepper, carrots, radish, onion	Chicken
S13—Indian lunch	Lettuce, lentils, tomato, dried tomato, celery, onion, sprouts and sunflower seeds, radish, sauce	-
S14—Fit	Cabbage, carrots, corn, peppers, fresh cucumber	Cheese, ham, Caesar sauce
S15—Tuna	Cabbage, sweet corn, pepper, canned peas	Mayonnaise, tuna paste
S16—Potato	Potatoes, cucumber, pickled peppers, onion, leek, spices	Mayonnaise
S17—Golden	Corn, peach, pineapple	Canned ham, mayonnaise
S18—Home	Carrots, potatoes, pickled peas, pickled cucumber, spices	Boiled egg, mayonnaise
S19—Athenian	Chinese cabbage, olives, canned peas, canned peppers, corn, vinaigrette	Feta cheese
S20—Classic	Chinese cabbage, canned peas, canned peppers, corn	Ham, cheese, mayonnaise
S21—Gyros	Chinese cabbage, pickled peas, pickled peppers, pineapple, corn	Chicken, gyros sauce
S22—Classic tuna	Chinese cabbage, canned peas, canned peppers, corn, horseradish sauce	Tuna
S23—Greek	Iceberg lettuce, cucumber, pepper, olives, onion, vinaigrette dressing	Feta cheese
S24—Balkan	Chinese cabbage, cucumber, pepper, carrots, onion	Feta, 1000 islands sauce
S25—Greek lunch	Iceberg lettuce, tomato, olives, arugula, cucumber, tomato, onion, sauce	Feta
S26—Fit oriental lunch	Iceberg lettuce, Chinese cabbage, Italian cabbage, lettuce, onion, pepper, carrots, pineapple, sunflower sprouts, rucola, sesame, vinaigrette sauce	Crab sticks
S27—Fit Italian lunch	Chinese cabbage, iceberg lettuce, cabbage, lettuce, onion, pepper, carrots, cucumber, corn, white bean, cherry tomato	Mozzarella cheese, yoghurt sauce
S28—Fit Greek lunch	Chinese cabbage, iceberg lettuce, tomatoes, white cabbage, onion, pepper, cucumber, corn, carrots, olives, red beans, spices, vinaigrette	Feta cheese
S29—Mexican	White cabbage, Chinese cabbage, carrots, corn, green peas, beans, spices	Mayonnaise
S30—Delecta	White cabbage, apples, leeks, carrots, spices	Mayonnaise

**Table 2 ijerph-19-01582-t002:** Nutritional value and other information about the salads.

Salad Symbol	Nutritional Value in 100 g of Product						Shelf Life [day]	Storage [°C]	Modified Atmosphere [+/−]	Preservatives [+/−]
	Energy [kJ/kcal]	Fat [g]	Saturates [g]	Sugars [g]	Protein [g]	Salt [g]				
S1	551/133	12.0	4.2	1.9	3.9	0.8	3	1–7	+	−
S2	419/100	5.0	1.1	6.1	7.2	1.2	5	1–7	+	+
S3	662/160	12.0	4.2	5.3	5.9	0.3	5	1–7	+	−
S4	397/96	6.6	0.6	2.8	5.5	0.7	1	1–6	+	+
S5	733/176	11.5	8.0	8.4	8.8	0.3	1	1–6	+	+
S6	334/80	3.4	1.4	5.9	5.4	1.8	1	1–6	+	−
S7	665/154	9.5	4.5	15.0	2.0	0.6	6	1–8	−	−
S8	253/61	3.0	0.7	4.4	2.5	0.6	2	1–6	−	−
S9	665/160	10.3	1.4	11.1	4.5	0.4	2	1–6	−	−
S10	470/113	8.3	1.9	5.0	3.4	0.8	2	1–6	−	−
S11	896/214	14.8	8.5	18.0	2.2	1.8	7	1–8	−	−
S12	439/105	6.3	0.8	4.5	7.2	0.3	2	1–8	−	−
S13	452/108	6.0	1.6	9.0	3.0	0.8	6	0–7	−	−
S14	419/100	4.5	3.5	8.4	6.5	1.5	6	2–7	−	+
S15	406/97	5.4	4.4	3.0	9.2	1.7	1	2–7	−	−
S16	389/93	3.5	0.4	14.5	1.0	1.2	1	0–7	−	−
S17	574/137	6.6	1.3	16.0	3.5	1.8	6	0–10	−	−
S18	389/93	4.6	0.3	11.8	1.2	1.2	3	0–4	−	−
S19	447/108	9.0	1.4	10.8	2.7	1.0	3	0–8	−	−
S20	461/111	8.4	1.9	5.9	4.6	0.7	2	1–8	−	−
S21	395/95	6.9	0.7	7.1	3.1	0.4	2	1–8	−	−
S22	345/83	5.5	0.5	12.2	4.5	0.5	2	1–8	−	−
S23	440/105	9.5	3.5	3.2	3.0	1.7	3	2–4	−	−
S24	276/66	5.1	0.8	4.5	2.0	1.2	3	2–4	−	−
S25	553/134	12.0	3.3	2.9	3.2	0.6	3	2–4	−	−
S26	528/126	9.1	4.6	3.3	7.6	0.9	3	0–8	−	−
S27	391/93	2.6	0.4	9.8	7.3	0.8	3	0–8	−	−
S28	645/154	13.5	4.1	4.9	1.6	1.1	3	0–8	−	−
S29	620/148	9.1	3.6	13.0	2.0	1.6	1	1–4	−	−
S30	590/141	9.6	1.1	11.1	1.8	0.3	1	1–4	−	−

Explanatory: (+)—modified atmosphere/preservatives presence; (−)—modified atmosphere/preservatives absence.

**Table 3 ijerph-19-01582-t003:** Microbiological quality of the tested RTE salads.

Salad Symbol	Count of Microorganisms [log CFU g^−1^]						Presence [+/−]	
	AC	STA	EB	EC	LAB	YM	SAL	LM
S1	5.00 ± 0.11	<1.00	3.91 ± 0.23	<1.00	8.43 ± 0.17	3.60 ± 0.00	+	−
S2	5.00 ± 0.00	<1.00	3.83 ± 0.01	<1.00	<1.00	4.32 ± 0.20	−	−
S3	6.98 ± 0.12	<1.00	4.04 ± 0.06	2.11 ± 0.01	7.10 ± 0.01	5.00 ± 0.00	+	−
S4	2.36 ± 0.06	<1.00	2.77 ± 0.13	<1.00	2.45 ± 0.10	2.00 ± 0.00	+	−
S5	3.89 ± 0.07	<1.00	2.70 ± 0.00	2.69 ± 0.02	3.26 ± 0.00	6.00 ± 0.00	−	−
S6	5.00 ± 0.00	<1.00	4.57 ± 0.02	5.00 ± 0.00	3.22 ± 0.00	2.42 ± 0.01	+	−
S7	3.26 ± 0.00	2.00 ± 0.00	6.32 ± 0.00	<1.00	4.48 ± 0.00	6.00 ± 0.00	−	+
S8	8.23 ± 0.16	<1.00	3.89 ± 0.00	4.08 ± 0.02	2.48 ± 0.00	6.00 ± 0.00	+	−
S9	7.91 ± 0.01	<1.00	6.83 ± 0.00	<1.00	4.69 ± 0.13	3.96 ± 0.08	−	+
S10	7.18 ± 0.00	<1.00	7.48 ± 0.12	<1.00	3.67 ± 0.27	6.00 ± 0.05	−	+
S11	7.76 ± 0.04	<1.00	5.00 ± 0.00	2.48 ± 0.12	1.86 ± 0.00	2.91 ± 0.19	−	−
S12	6.32 ± 0.00	<1.00	5.00 ± 0.00	2.99 ± 0.05	1.00 ± 0.00	5.00 ± 0.00	−	+
S13	6.52 ± 0.00	<1.00	6.32 ± 0.12	2.74 ± 0.24	1.00 ± 0.00	3.11 ± 0.00	−	+
S14	6.52 ± 0.00	<1.00	6.00 ± 0.00	4.08 ± 0.00	2.51 ± 0.19	2.00 ± 0.00	−	−
S15	6.66 ± 0.10	<1.00	6.36 ± 0.14	2.90 ± 0.00	2.33 ± 0.00	2.00 ± 0.02	−	+
S16	8.64 ± 0.06	<1.00	<1.00	<1.00	5.0 ± 0.00	7.00 ± 0.67	+	+
S17	4.95 ± 0.01	3.54 ± 0.00	2.95 ± 0.10	<1.00	<1.00	2.00 ± 0.00	+	+
S18	4.04 ± 0.00	2.91 ± 0.03	<1.00	<1.00	<1.00	1.00 ± 0.00	−	−
S19	8.12 ± 0.58	<1.00	2.08 ± 0.00	1.80 ± 0.00	6.60 ± 0.02	2.30 ± 0.00	−	−
S20	6.54 ± 0.00	<1.00	1.30 ± 0.05	1.00 ± 0.00	1.30 ± 0.05	<1.00	−	−
S21	4.45 ± 0.05	<1.00	<1.00	<1.00	<1.00	4.30 ± 0.02	−	−
S22	6.30 ± 0.00	<1.00	<1.00	<1.00	1.55 ± 0.00	3.14 ± 0.08	−	+
S23	9.30 ± 0.20	<1.00	6.84 ± 0.50	5.55 ± 0.05	7.80 ± 0.50	6.60 ± 0.00	−	−
S24	5.12 ± 0.43	<1.00	3.12 ± 0.03	2.30 ± 0.01	4.40 ± 0.03	3.01 ± 0.06	+	−
S25	7.90 ± 0.00	<1.00	2.34 ± 0.00	1.22 ± 0.08	5.90 ± 0.04	5.80 ± 0.00	−	−
S26	6.86 ± 0.00	<1.00	5.00 ± 0.12	<1.00	3.70 ± 0.10	2.12 ± 0.04	−	−
S27	5.30 ± 0.10	<1.00	4.14 ± 0.02	<1.00	2.60 ± 0.00	<1.00	−	−
S28	6.99 ± 0.01	<1.00	<1.00	<1.00	5.63 ± 0.07	4.38 ± 0.14	−	+
S29	7.24 ± 0.00	<1.00	<1.00	<1.00	2.45 ± 0.30	4.16 ± 0.02	−	−
S30	6.00 ± 0.00	<1.00	4.38 ± 0.01	2.90 ± 0.12	1.90 ± 0.00	1.18 ± 0.03	−	−

Explanatory: AC—aerobic mesophilic total flora, STA—*S. aureus*, EB—*Enterobacteriaceae* family, EC—*E. coli*, LAB—lactic acid bacteria, YM—yeasts and molds; SAL—*Salmonella* spp., LM—*L. monocytogenes*; <1.00—below the detection level. The results are shown as log CFU g^−1^; means ± standard deviation; (+) presence of bacteria in 25 g of the product, (−) absence of bacteria in 25 g of the product; n = 3.

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
