# Peer review of "Assessment of the Microbiological Quality of Ready-to-Eat Salads—Are There Any Reasons for Concern about Public Health?"

_ijerph, 2022, doi:10.3390/ijerph19031582_

Round 1
Reviewer 1 Report
Lepecka and co-workers presented a well written manuscript with plenty of information the public health and the incidence of food-borne disease. There is sufficient information the introduction. The authors also presented details of the materials and methods for others to replicate the study. The tables of the results are clearly presented. I had a bit of trouble understanding both Figures 1 and 2. I wonder if there are other ways to present the same results so it is a lot more obvious to the average reader. The discussion and conclusions are easy to follow and well written. I only found very few grammatical punctuation errors.
All the best with this manuscript and acceptance for publication.
Author Response
I would like to thank the Reviewer for his comments on the manuscript.
Reviewer 2 Report
The background of this study is the increasing interest of consumers in ready-to-eat food products that, not requiring further preparation, are a convenient solution for busy consumers. The authors investigated the microbiological quality of 30 ready-to-eat salads, available on Polish markets, through strict samples preparation and microbiological and statistical analysis. They found, in most cases, the presence of microorganisms (bacteria, yeasts, and molds), sometimes pathogenic, arguing that the microbiological quality of the evaluated products was not satisfactory. They concluded that further investigations, possibly with a larger amount of samples of different types, in terms of packaging method and season of production, would provide information of great interest to both the legislators and producers, encouraging manufacturers to improve the quality of these kinds of food. The matter of study is of great interest to public health, the expression was clear and the content of the review well explained. I strongly recommend accepting it in this form.
Author Response

(The authors gave the same response as above.)

Reviewer 3 Report
Review for the manuscript entitled Assessment of the microbiological quality of ready-to-eat salads - are there any reasons for concern about public health? (ijerph-1516323)
This manuscript presents an interesting study about the microbiological quality of ready to eat salads. The study is important not only for the producers, but for consumers and legislators also. The study has a sound research topic, objectives and experimental design. Overall, the manuscript is well written. I have only minor suggestions which are more connected to the form of the article and not the content. In conclusion, the manuscript can be improved prior to publishing and in my opinion, after some minor revisions it can be published in ijerph.
Here are some suggestions for the Authors:
L45-46 Please revise “It is estimated that in Europe in 2021… “ 2021 has ended already. Update the reference or reedit the phrase.
L108 Authors claim that “The products were fresh, before the expiry date, …” Did you try to correlate the microbiological quality with the number of days since the salad was produces/packaged?
L125-126 Authors claim that “25 g of the salad product was weighed by taking different ingredients to obtain a representative sample”. Usually in a salad, each ingredient has different yields. It is not clear if the “representative sample” contained the same ratio of the ingredients as in the original salad.
L145: Please provide the apparatus details: producer, model, city, country (incubator).
L146, L159: “Then, 1 mL was withdrawn…” might be rephrased as “An aliquot of 1 ml…”
Author Response
I would like to thank the Reviewer for preparing the manuscript review. All comments were carefully analyzed and the manuscript was revised as indicated.
Line 46-47. Thank you for your suggestion. The sentence has been converted into the past tense.
Line 108. Explanation: The ready-to-eat salads that we purchased for testing were delivered to the commercial network up to 1 day before the test. The label showed the batch numbers of the product and the date of manufacture, which indicated that it was delivered to the store in less than 24 hours. In this study, we assumed that since salads are 1 day old (from production), we consider them fresh. In addition, the salads were immediately inspected on the day of purchase to avoid possible bacterial and mold growth due to longer storage of the product (regardless of the fact that salads should maintain an appropriate microbiological quality throughout their shelf-life).
Line 125-126. We agree with the reviewer's opinion. However, salads are a heterogeneous product and the proportion of ingredients may vary depending on the batch or package, especially since a mix of salads is used in many products. We tried to select the ingredients of the salads so that they were homogeneous and could be considered representative.
Line 138-139. Incubator name and manufacturer data have been added.
Line 146. The sentence was converted according to the Reviewer's remark: “An aliquot of 1 mL was withdrawn and transferred to 9 mL of broth and incubated again at 37 °C for 24 h.”
We hope that our corrections are sufficient to accept the manuscript for further publication stages.